# *Bombyx mori* Ecdysone Receptor B1 May Inhibit BmNPV Infection by Triggering Apoptosis

**DOI:** 10.3390/insects14060505

**Published:** 2023-05-31

**Authors:** Zhihao Su, Chunxiao Zhao, Xinming Huang, Junli Lv, Ziqin Zhao, Kaiyi Zheng, Xia Sun, Sheng Qin, Xueyang Wang, Byung-Rae Jin, Yangchun Wu

**Affiliations:** 1Jiangsu Key Laboratory of Sericultural Biology and Biotechnology, School of Biotechnology, Jiangsu University of Science and Technology, Zhenjiang 212100, China; 2The Key Laboratory of Silkworm and Mulberry Genetic Improvement, Ministry of Agriculture, Sericultural Research Institute, Chinese Academy of Agricultural Science, Zhenjiang 212100, China; 3College of Natural Resources and Life Science, Dong-A University, Busan 49315, Republic of Korea

**Keywords:** *Bombyx mori*, *BmEcR-B1*, BmNPV, 20E, apoptosis

## Abstract

**Simple Summary:**

Ecdysone is an essential factor affecting host immunity, in addition to regulating insect metamorphosis. There are two isoforms of ecdysone receptors (EcR) in silkworm, BmEcR-B1, and BmEcR-A. In this study, *BmEcR-B1* was found to be more critical than BmEcR-A in responding to BmNPV infection. Further studies confirmed that *BmEcR-B1* played an anti-BmNPV role in the participation of 20E, which was associated with 20E-induced apoptosis. This study provides theoretical support for elucidating the resistant mechanism of silkworm to BmNPV and the control of BmNPV disease in sericulture.

**Abstract:**

Bombyx mori nucleopolyhedrovirus (BmNPV) is a serious threat to sericulture. Nevertheless, no effective control strategy is currently available. The innate immunity of silkworm is critical in the antiviral process. Exploring its molecular mechanism provides theoretical support for the prevention and treatment of BmNPV. Insect hormone receptors play an essential role in regulating host immunity. We found a correlation between *Bombyx mori ecdysone receptor B1* (*BmEcR-B1*) and BmNPV infection, whereas the underlying mechanism remains unclear. In this study, the expression patterns and sequence characteristics of *BmEcR-B1* and its isoform, *BmEcR-A*, were initially analyzed. *BmEcR-B1* was found to be more critical than *BmEcR-A* in silkworm development and responses to BmNPV. Moreover, RNAi and an overexpression in BmN cells showed *BmEcR-B1* had antiviral effects in the presence of 20-hydroxyecdysone (20E); Otherwise, it had no antiviral activity. Furthermore, *BmEcR-B1* was required for 20E-induced apoptosis, which significantly suppressed virus infection. Finally, feeding 20E had no significant negative impacts on larval growth and the cocoon shell, suggesting the regulation of this pathway has practical value in controlling BmNPV in sericulture. The findings of this study provide important theoretical support for understanding the mechanism of the silkworm innate immune system in response to BmNPV infection.

## 1. Introduction

Silkworm is not only a kind of economic insect, but it is also widely used in scientific studies as a model organism of Lepidoptera. Sericulture is constantly threatened by BmNPV, which generates huge economic losses each year [1]. However, there is still no effective way to solve this problem. One essential method to solving the current problem is understanding the antiviral molecular mechanism of silkworms. This will provide a theoretical basis for breeding resistant silkworm strains and solving the viral disease in sericulture [2]. However, the antiviral mechanism of silkworm against BmNPV is quite complex, and the current research data are insufficient to define it. As a result, the identification and study of BmNPV-associated genes or proteins is a pressing issue that must be addressed.

Insects, the most abundant group, have evolved a robust immune system to resist disease invasion. The steroid hormone ecdysone can induce apoptosis, promoting developmental and physiological activities, including molting, metamorphosis, and reproduction [3,4]. In addition, ecdysone can also promote synthesis, such as the biosynthesis of chitin [5]. Apoptosis, a type of cellular immunity, is essential not just for average body growth but also for resisting pathogen infections [6,7]. Studies on BmNPV-infected cells showed that apoptosis was significantly activated, which significantly eliminated or restricted the formation of progeny viruses [8,9]. These findings indicated the crucial role of apoptosis in resistance to viral infections. Ecdysone is an apoptosis-regulating factor [10], and EcR is essential in ecdysone-signaling transduction [11]. However, whether the EcR-mediated apoptosis pathway is involved in response to BmNPV infection has not yet been reported.

The ecdysone receptor is a non-covalent heterodimer of two proteins, EcR and ultraspiracle (USP) [12]. The heterodimer is bound and subsequently activated to regulate target genes that elicit proliferation, cell death, and differentiation [13]. It has been reported that EcR contains three isoforms in *Drosophila melanogaster*, A, B1, and B2, whereas Lepidopterans have two isoforms, A and B1 [14]. In silkworms, the expression patterns of EcR-A and EcR-B1 are different. *BmEcR-A* is predominantly expressed in the anterior silk glands (ASGs) at the beginning of metamorphosis, while *BmEcR-B1* is expressed in various tissues [15]. In silkworm ASGs, *BmEcR-B1* is important for initiating apoptosis and autophagy [16]. In silkworm eggs, BmEcR-B1 can promote embryonic development by binding to 20E [17].

In this study, the expression levels of *BmEcR-B1* and *BmEcR-A* were analyzed in different tissues of different resistant silkworm strains, revealing that *BmEcR-B1* may have broader functions than *BmEcR-A* in silkworm development and responses to BmNPV infection. Furthermore, the function of *BmEcR-B1* in BmNPV infection was confirmed by RNAi and the overexpression system. Moreover, the role of *BmEcR-B1* in 20E-induced apoptosis in response to BmNPV infection was determined. The results provided a theoretical basis for understanding the silkworm’s antivirus mechanism and antiviral molecular breeding.

## 2. Materials and Methods

### 2.1. Silkworms and BmN Cells

Silkworm strains, YeA, YeB and p50 (Dazao), were maintained in the Key Laboratory of Sericulture, College of Life Sciences, Jiangsu University of Science and Technology, Zhenjiang, China. The first three instars of larvae were fed fresh mulberry leaves at 26 ± 1 °C and relative humidity of 75 ± 5% on a 12-h daily cycle, whereas the final two instars were fed at 24 ± 1 °C, and other conditions remained unchanged.

A BmN cell line derived from the silkworm ovary was cultured in TC-100 (AppliChem, Darmstade, Germany) medium supplemented with 10% (*v*/*v*) fetal bovine serum (FBS) (Thermo Fisher Scientific, New York, USA) and 1% (*v*/*v*) penicillin and streptomycin (AppliChem, Darmstade, Germany) at 28 °C.

### 2.2. Preparation and Detection of BmNPV

BV-eGFP was maintained in our laboratory and amplified in BmN cells. The eGFP gene was inserted between *BamH*I and *Xho*I and activated expression with a polyhedral promoter. The titer of BV-eGFP (pfu/mL) was calculated using the standard curve method according to our previous study [18]. A volume of 2.0 μL of BV-eGFP (1 × 10^8^ pfu/mL) was used to inject silkworm larvae using capillary, and 20 μL was used to inoculate BmN cells. The eGFP signal was captured with an inverted microscope DMi3000B camera (Leica, Solms, Germany), and pictures were processed with Application Suite V4.6 software (Leica, Solms, Germany). Occlusion-derived virus (ODV) was harvested from infected silkworm hemolymph, and the purified ODV was dissolved in double distilled water (ddH_2_O). The density of ODV was determined by a hemocytometer.

### 2.3. Sample Preparation for Expression Pattern Analysis

To determine the spatio-temporal expression pattern of *BmEcR-A* and *BmEcR-B1*, eggs at different stages, tissues on the third day of fifth instar larvae, and the whole body at first, second, third, fourth, fifth, pupa, and adult of p50 were collected. To analyze the immune response of *BmEcR-A* and *BmEcR-B1* to BmNPV, different tissues of YeA and YeB on the third day of fifth instar were collected after being fed with 5 μL of ODV (1.0 × 10^5^ OB/mL) for 48 h, including the Malpighian tube, hemolymph, fat body, and midgut. The ddH_2_O treatment was used as a blank control. Thirty larvae or tissues were mixed in each group to minimize individual genetic differences, and each group was repeated three times. All samples were quickly powdered with liquid nitrogen and stored at −80 °C until use.

### 2.4. Bioinformatics Analysis

*BmEcR-A* (ID: NM_001173377.1), *BmEcR-B1* (ID: NM_001173375.1), and their homologs in other species (Appendix A) were obtained from the National Center for Biotechnology Information (NCBI, http://www.ncbi.nlm.nih.gov/, accessed on 23 October 2022). The functional domain was predicted using the online SMART server (http://smart.embl-heidelberg.de/, accessed on 15 February 2020). The sequences homology of EcR-A and EcR-B1 amino acid sequence in different species were analyzed using DNAMAN 8.0 software. MEGA 7.0 software was used to generate the evolutionary relationship of EcR-A and EcR-B1 in different species using the Neighbor–Joining method with 1000 bootstrap replications and an optimal DNA/protein model of LG + G. The evolutionary distances were computed using the Maximum Composite Likelihood method.

### 2.5. RNA Extraction and the First Strand cDNA Synthesis

The total RNA was extracted using RNAiso Plus (TaKaRa Biotechnology Co., Ltd., Dalian, China) according to the manufacturer’s instructions. RNA precipitate was dissolved in RNase-free ddH_2_O after cleaning with 75% ethanol. The concentration and purity of RNA were measured by a NanoDrop 2000 spectrophotometer (Thermo Fisher Scientific, New York, NY, USA). The integrity of RNA was determined by 1% agarose gel denaturation electrophoresis. The qualified RNA was stored at −80 °C until use. The cDNA was generated using the PrimeScript^TM^ RT Reagent Kit (TaKaRa Biotechnology Co., Ltd., Dalian, China; with gDNA Eraser) following the manufacturer’s instructions. The qualified cDNA was stored at −20 °C.

### 2.6. Real-Time Quantitative PCR (RT-qPCR)

The relative expression levels of genes were determined using RT-qPCR. Table 1 shows all primers that were generated on the NCBI website. A total of 10 μL of RT-qPCR reaction system consisted of 5.0 μL of 2 × NovoStart^®^ SYBR qPCR SuperMix Plus (Novoprotein, Suzhou, China), 0.5 μL of upstream and downstream primers, 1.0 μL of template, and 3.0 μL of ddH_2_O. The reaction was performed on the LightCycler^®^ 96 system (Roche, Basel, Switzerland) using pre-denaturation at 95 °C for 30 s, followed by 40 cycles at 95 °C for 20 s and 60 °C for 60 s. The raw data were processed with the LightCyler 96 SW 1.1 software (Roche, Basel, Switzerland). All samples were replicated three times, and the relative expression levels were calculated using the 2^−ΔΔCT^ method. The optimal reference gene of *Bombyx mori glyceraldehyde-3-phosphate dehydrogenase* (*BmGAPDH*) was selected based on a previous report [19].

### 2.7. Synthesis of siRNA and Transfection

To knock down the expression of *BmEcR-B1* in BmN cells, two targets located in the function domain of *BmEcR-B1* were designed on the online website (https://www.genscript.com/tools/sirna-target-finder accessed on 23 October 2022). siRNAs targeting *BmEcR-B1* functional domain are named siEcR-B1. siRNA targeting red fluorescent protein (RFP) is named siRFP. The siRNA used in this study was short RNA of 19 bp. Table 2 shows primers with target DNA sequences placed behind the T7 promoter. The siRNA was synthesized using oligo primers by the In Vitro Transcription T7 Kit (for siRNA synthesis, TaKaRa Biotechnology Co., Ltd., Dalian, China) according to the manufacturer’s instructions. Briefly, Olig-1 to 4 were used to synthesize one siRNA product. Olig-1/2 and Olig-3/4 were used to synthesize the forward and reverse templates required to transcribe siRNA, and the two strands were annealed separately and then hybridized. The quality of siRNA was confirmed by 3% agarose gel electrophoresis. The purity and concentration were determined by the NanoDrop 2000 spectrophotometer (Thermo Fisher Scientific, New York, NY, USA). The qualified siRNA was stored at −80 °C until use.

The transfection complex was prepared using a Neofect^TM^ DNA transfection reagent (NEOFECT, Beijing, China) according to the manufacturer’s instructions. Briefly, 4 μg of each siRNA was gently mixed with 200 μL of serum-free TC-100 medium, followed by 4 μL of Neofect^TM^ Transfection reagent that were thoroughly mixed with 200 μL of serum-free TC-100 medium and then maintained at room temperature for 30 min. The transfection complex was then applied to the prepared BmN cells and cultured at 28 °C, and the effect of siRNA on target genes was determined after 24 h of transfection using RT-qPCR.

### 2.8. Construction of the Overexpression Vector

The functional domain of *BmEcR-B1* was amplified with the primes BmEcR-B1-OP (Table 1; the underlined portions indicate the *EcoR*I and *Xba*I restriction sites, respectively). The purified PCR products were cloned into the pMD19-T vector for sequencing. The confirmed pMD-19T-BmEcR-B1 and the pIZT/V5-His-mCherry were digested with *EcoR*I and *Xba*I and then ligated for 12 h at 16 °C using T4 DNA ligase (TaKaRa Biotechnology Co., Ltd., Dalian, China). The recombinant plasmid pIZT/V5-His-mCherry-BmEcR-B1 was validated by the two restriction enzymes and then sequenced at Sangong Biotech (Shanghai, China) to ensure its accuracy. The original vector of pIZT/V5-His-mCherry was set as a negative control. Transfection of overexpression vectors is the same as for siRNA.

### 2.9. The Treatment of 20E on Silkworm Larvae

Five series concentrations of 20E dissolved in ddH_2_O were used to identify the optimal dose for activating *BmEcR-B1*, including 0, 13, 45, 130, 260, and 390 μg/mL. To detect the effect of 20E on *BmEcR-B1* expression and BmNPV reproduction, 4th instar larvae were given the optimal dose of 20E after being inoculated with 2 μL of BV-eGFP for 24 h, 48 h, and 72 h.

To determine the effect of 20E on silkworm resistance to BmNPV, 540 larvae of p50 on the first day of second instar were divided into six groups, each with three equal repetitions. Each group was fed 1 × 10^6^ OB/mL ODV-treated mulberry leaves, and then the larvae were fed with the optimal dose of 20E at 0.5 d, 1.0 d, 1.5 d, and 2.0 d for half a day after BmNPV infection. Fresh mulberry leaves (1 cm^2^) were soaked in different concentrations of 20E solution, dried naturally, and fed to silkworm larvae until the leaves were consumed, and then replaced with fresh mulberry leaves. The mortality of third instar silkworm larvae was recorded every 12 h after infection.

To study the effects of 20E on the growth of silkworms, the weight of silkworm larvae at different developmental stages, as well as the weight of cocoon shells, were counted after being fed with the optimal dose of 20E at 0.5 d, 1.0 d, 1.5 d, and 2.0 d after the first day of second instar. Thirty silkworm larvae at second, third, fourth, and fifth instar were weighed independently in each treatment group, and 30 cocoons of each sex were weighed independently. Then, the average weight of each larva or cocoon shell was calculated.

### 2.10. The Effect of 20E Treatment on BmEcR-B1 Expression and BmNPV Infection in BmN Cells

The optimal dose of 20E was coupled with 4 μg of siEcR-B1, which was subsequently introduced into BmN cells. siRFP was set as the control. *BmEcR-B1* expression and BmNPV infection were detected after 48 h of treatment using RT-qPCR and fluorescence microscope. The same treatment was repeated in overexpression group, with the difference that BmN cells contained pIZT/V5-His-mCherry-BmEcR-B1 was treated directly with 20E.

### 2.11. Caspase Activity Assay and Apoptotic Bodies Detection

The 20E/siEcR-B1-treated group was set as the recovery experiment of the 20E-treated group. The control was 20E/siRFP. The changes in caspase activity were detected using the caspase-3 activity assay test kit (Njjcbio, Nanjing, China) according to the manufacturer’s instructions. BmN cells were treated with lysis buffer containing DTT on ice for 30 min. The supernatant was collected and incubated with the reaction solution containing DTT and Ac-DEVD-pNA at 37 °C overnight. The absorbance value was detected by Spectramax i3 multifunctional microplate reader (Molecular Devices, CA, USA), and the degree of caspase activation was generated based on the absorbance value. Apoptotic bodies were detected by the TUNEL apoptosis detection kit (YEASEN, Shanghai, China) according to the manufacturer’s instructions. BmN cells were collected and resuspended in 1 × phosphate-buffered saline (PBS). Each cell smear was incubated with an equilibration buffer at room temperature for 30 min and then mixed with terminal deoxynucleotidyl transferase (TdT) incubation buffer at 37 °C for 1 h in the dark. DAPI (Sangon Biotech, Shanghai, China) was used to stain the cell nucleus. The fluorescence signal was detected by the OLYMPUS IX3 inverted fluorescence microscope (OLYMPUS, Tokyo, Japan), and the captured data of the control group setup is used as the background.

### 2.12. Statistics Analysis

Data statistical analysis and charting were completed in GraphPad Prism 8 software (GraphPad Software, San Diego, CA, USA). The SPSS Statistics 20 software (IBM, Endicott, New York, NY, USA) was used to assess differences between samples using one-way ANOVA. The Kruskal Wallis test was performed to analyze the data that did not match the normality in SPSS software. Student’s *t*-test was performed to detect the significance of the difference between the two data sets that follow the normal distribution. *p* < 0.05 was considered statistically significant.

## 3. Results

### 3.1. Bioinformatics Analysis

The amino acid sequences of BmEcR-A and BmEcR-B1 and their homologs in other species were aligned using DNAMAN 8.0 software (Appendix A). The results showed that the amino acid sequence of the ZnF C4 domain in both BmEcR-A and BmEcR-B1 are conserved, while the HOLI domain is absent in BmEcR-A. Moreover, the evolutionary relationship of EcR-A and EcR-B1 in different species was clustered into two separate groups (Appendix A). These results suggested that the function of BmEcR-B1 might be different from that of BmEcR-A. Furthermore, BmEcR-B1 showed a closer evolutionary relationship with *Bombyx mandarina*, *Galleria mellonella*, *Manduca sexta*, and *Antherae pemyi* than other Lepidopterans, suggesting that *EcR-B1* might be genetically distinct between Lepidopterans.

### 3.2. The Temporal and Spatial Expression Profile of BmEcR-B1 and BmEcR-A

The p50 strain has been used to build the silkworm genome [20]; detecting expression patterns in p50 would provide a better data reference. Thus, the expression profiles of *BmEcR-A* and *BmEcR-B1* in different developmental stages and tissues of p50 were analyzed using RT-qPCR. The results showed that both the expression of *BmEcR-A* and *BmEcR-B1* were higher in the early developmental stages of eggs than that in the late stages (Figure 1A). Their expression levels in the testis and ovary were the highest among various tissues (Figure 1B). Relatively high expression levels of *BmEcR-A* and *BmEcR-B1* were found during metamorphosis, including before molting of fourth instar, pupa, and adult (Figure 1C). Although *BmEcR-A* shared a similar expression pattern with *BmEcR-B1* in different developmental stages and different tissues, the expression level of *BmEcR-A* was significantly lower than that of *BmEcR-B1*.

### 3.3. BmEcR-B1 Showed a Significant Response to BmNPV Infection

To preliminarily compare the role of *BmEcR-B1* and *BmEcR-A* in response to BmNPV, their relative expression levels in four immune-related tissues of YeA (resistant strain, LC_50_ > 10^9^ OB/mL) and YeB (susceptible strain, LC_50_ = 10^5^ OB/mL) after BmNPV infection at 48 h were analyzed using RT-qPCR. The results showed that the immune response of *BmEcR-B1* to BmNPV was much more significant than that of *BmEcR-A* in four tissues, except in the hemolymph (Figure 2A). In addition, *BmEcR-B1* expression was significantly higher in YeB tissues after BmNPV infection, except for the Malpighian tube (Figure 2D) and markedly lower in YeA tissues after infection, except for the midgut (Figure 2C). The significantly different expression levels of *BmEcR-B1* in the two strains after BmNPV infection indicated that it might be the key gene in response to viral infection.

### 3.4. E activates BmEcR-B1 Expression and Inhibits BmNPV Infection In Vivo

EcR is a 20E receptor that mediates 20E-regulated host immune response [21], while its role in silkworm remains unclear. To analyze the role of 20E on *BmEcR-B1* expression and BmNPV infection, the relative expression levels of *BmEcR-B1* and viral capsid gene *vp39* were determined after 24 h, 36 h, and 48 h of 20E treatment, as well as the survival rate of infected larvae. The results showed that 45 μg/mL of 20E were the most effective for activating *BmEcR-B1* expression (Figure 3A). *BmEcR-B1* expression was significantly increased at 36 h and 48 h after 45 μg/mL of 20E treatment (Figure 3B). The survival rate of the third instar larvae was significantly increased in the 20E-treated groups, and the effect was most pronounced after 0.5 days of 20E treatment compared with other time points (Figure 3C). The expression of *vp39* decreased significantly at 36 h and 48 h after 20E treatment (Figure 3D). Therefore, 20E has an important role in regulating *BmEcR-B1* expression and inhibiting BmNPV infection.

### 3.5. BmEcR-B1 Plays a Vital Role in 20E Inhibition of BmNPV In Vitro

To further analyze the relationship of 20E, *BmEcR-B1*, BmNPV, and BmN cells were selected for in-depth analysis. The regulation effects of 20E on *BmEcR-B1* expression and BmNPV replication were analyzed again in BmN cells, and the results (Figure 4A–C) were consistent with the above in-vivo results (Figure 3), confirming that 20E plays an important role in activating *BmEcR-B1* expression and inhibiting viral infection. On this basis, two experiments were designed to assess the involvement of *BmEcR-B1* in the inhibition of BmNPV infection by 20E. The first was the knockdown of *BmEcR-B1* with siEcR-B1 in the presence of 20E (Figure 4D–F), and the other one was in the absence of 20E (Figure 4G–H). During the presence of 20E, the inhibition of 20E on BmNPV titer and capsid gene *vp39* was eliminated after the knockdown of *BmEcR-B1* (Figure 4D–F), while no significant difference was observed in the absence of 20E (Figure 4G–H). The results showed that *BmEcR-B1* is vital for 20E to inhibit viral proliferation, and 20E is also required for *BmEcR-B1* resistance to BmNPV.

### 3.6. Overexpression of BmEcR-B1 Inhibits BmNPV Infection in BmN Cells

To further validate the RNAi results that knockdown *BmEcR-B1* benefits BmNPV infection, *BmEcR-B1* was overexpressed in BmN cells using pIZT-mCherry-BmEcR-B1. The results of correct BmEcR-B1 fragmentation (Figure 5D), red fluorescent signals of mCherry (Figure 5A–C), and significantly upregulated levels of *BmEcR-B1* indicated that it was overexpressed in BmN cells. Moreover, an extremely high expression in the 20E-treated group revealed that it could be regulated by 20E (Figure 5E).

To analyze the effect of *BmEcR-B1* overexpression on viral infection, eGFP intensity, *vp39* expression, and virus titer were analyzed in different groups at 24 h, 48 h, and 72 h (Figure 6). The results showed that BmNPV proliferation was significantly decreased in *BmEcR-B1* overexpression group (Figure 6A,B,D), and it was even lower after 20E treatment (Figure 6C,E). When these findings are combined with the results in Figure 4A–C, it is reasonable to assume that *BmEcR-B1* plays an important role in limiting viral replication in the presence of 20E.

### 3.7. Inhibition of Viral Infection by BmEcR-B1 Is Associated with the Activation of Apoptosis

To analyze whether BmEcR-B1 inhibits BV-eGFP infection via the activation of apoptosis, apoptosis bodies and caspase activity were detected in the 20E-treated and 20E/siEcR-B1-treated groups. The 20E/siEcR-B1-treated group was set as the recovery experiment of the 20E-treated group. The results showed that the quantity of apoptotic bodies was increased after 20E induction, while it was recovered by siEcR-B1 (Figure 7A, B). Moreover, Caspase activities were increased in the 20E-treated group (Figure 7C), while it also reduced significantly after siEcR-B1 transfection (Figure 7D). The results suggested the inhibition of BmNPV infection by *BmEcR-B1* is associated with the activation of apoptosis.

### 3.8. The Analysis of the Effect of 20E on Silkworm Larvae Development and Cocoon Shell

To analyze whether 20E can be used to control BmNPV in sericulture, the effect of 45 μg/mL of 20E on silkworm larvae development and cocoon score was evaluated. Silkworm larvae on the second instar were fed with 20E at 0.5 d, 1.0 d, 1.5 d, and 2.0 d after ecdysis, respectively. The larval weight of the silkworm was recorded at second, third, fourth, and fifth instars, as well as the cocoon shell weight. The results showed that feeding 20E reduced larval weight at different development stages and cocoon shell weights compared to the control group, but the differences were not statistically significant (Figure 8A,B).

## 4. Discussion

*BmEcR-B1* was found to be significantly differentially expressed following BmNPV infection in our earlier study [22]. Related studies have revealed that EcR contributes to insect immune response [23], but it is uncertain whether *BmEcR-B1* also has an immunological function in silkworms. In this study, the importance of *BmEcR-B1* and *BmEcR-A* was first compared, and then the function of *BmEcR-B1* was determined in vivo and in vitro.

### 4.1. BmEcR-B1 Is More Critical Than BmEcR-A in Silkworm

There are two kinds of isoforms of EcR in silkworms, BmEcR-B1, and BmEcR-A. Although the expression patterns of BmEcR-B1 and BmEcR-A have been reported [15], their sequence characteristics are still unclear. In this study, the differences in the expression between *BmEcR-A* and *BmEcR-B1* were analyzed in different developmental stages and tissues. Although *BmEcR-A* shared a similar expression pattern with *BmEcR-B1*, the relative expression level of *BmEcR-A* was significantly lower than *BmEcR-B1* (Figure 1). Moreover, *BmEcR-A* lacked the HOLI domain compared to *BmEcR-B1* (Appendix A). HOLI domain is a ligand-binding domain. It has been reported that EcR-A, which lacks the HOLI domain in *Drosophila*, completely loses its activation of ecdysone-induced genes even in the transgenic expression of EcR-A [24]. Thus, the higher expression of *BmEcR-B1* suggested that it has broader functions than *BmEcR-A* during silkworm development. Moreover, the immune response of *BmEcR-B1* was more significant than that of *BmEcR-A* after BmNPV infection (Figure 2), suggesting that *BmEcR-B1* might be a key gene in silkworm in response to BmNPV infection but not *BmEcR-A*.

### 4.2. BmEcR-B1 Plays a Vital Role in Inhibiting BmNPV, Which Needs the Presence of 20E

It was reported that EcR belongs to the nuclear receptor superfamily of 20E, which requires the activation of 20E [25], but its role in silkworm has not been reported yet. In this study, the relationship between 20E and BmEcR-B1 was determined. We found that 20E could activate the expression of *BmEcR-B1*, and the optimal concentration was 45 μg/mL (Figure 3A,B). One of the previous studies used 10 mg/mL of 20E to inject silkworms to regulate gene expression [26]. Thus, it should be reasonable that 45 mg/mL of 20E was the optimal concentration to upregulate *BmEcR-B1* expression. Moreover, 20E has been reported to regulate the silkworm innate immune [27], with the exception of BmNPV. This study found that feeding 20E enhanced silkworm resistance to BmNPV infection and inhibited *vp39* expression (Figure 3C,D), suggesting that 20E had an important role in silkworm immunity of silkworm to BmNPV. To determine the role of *BmEcR-B1* in 20E-induced inhibition against BmNPV, siRNA targeting on *BmEcR-B1* was used to knock it down. We found that the knockdown of *BmEcR-B1* increased BmNPV infection in the presence of 20E (Figure 4D–F), but without effect in the absence of 20E (Figure 4G–I). EcR was reported to be silent in the absence of 20E [26], which may be the reason why the knockdown of *BmEcR-B1* also did not affect viral infection. To further confirm the function of *BmEcR-B1* in BmNPV infection, it was overexpressed using pIZT/V5-mCherry vector in BmN cells (Figure 5). It was found that BmNPV infection was significantly inhibited after overexpression of *BmEcR-B1* (Figure 6D,F), and the effect was better in the presence of 20E (Figure 6E,G). Thus, *BmEcR-B1* was proved to play a vital role in the resistance to BmNPV infection, and this requires the participation of 20E. When the function of EcR in Dengue Virus 2 [28] and *Helicoverpa armigera* SNPV [29] is considered, it is plausible to conclude that EcR is a vital gene for host immunity to the virus.

### 4.3. BmEcR-B1 Likely Inhibits Virus Infection in BmN Cells via the Activation of Apoptosis

Although *BmEcR-B1* has been shown to have an anti-BmNPV effect, the underlying mechanism remains unclear. Studies have shown that 20E and *BmEcR-B1* can activate apoptosis [30,31,32]. To analyze whether the anti-BmNPV role of *BmEcR-B1* is related to this, the present study analyzed the apoptotic bodies and caspase activity in 20E treated-group after RNAi of *BmEcR-B1*. The significant increase in apoptotic bodies and caspase activities in the 20E-treated group (Figure 7B,C), whereas this activation effect was eliminated after siEcR-B1 treatment (Figure 7D), suggesting that *BmEcR-B1* inhibited BmNPV via the activation of apoptosis. The results provide another example of the association between *BmEcR-B1* and apoptosis. Moreover, it was reported that 20E-induced apoptosis was inhibited when the binding of USP and EcR-B1 was blocked by silkworm 30K proteins [33]. Thus, the inhibition of BmNPV by *BmEcR-B1* is related to the activation of apoptosis, which provides a vital supplement for EcR-mediated 20E triggering apoptosis in response to viral infection. Finally, to evaluate whether 20E can enhance silkworm antivirus levels, the larval weight and cocoon shell weight of silkworms at different developmental stages were determined after feeding with 20E at different times, and the results showed insignificant impacts, making it feasible (Figure 8).

In conclusion, the activation of *BmEcR-B1* plays an important role in inhibiting BmNPV infection by activating apoptosis, which needs the existence of 20E (Figure 9). Moreover, as 20E has an insignificant effect on silkworm development and cocoons, it can be used in sericulture to reduce the losses caused by BmNPV.

## Figures and Tables

**Figure 1 insects-14-00505-f001:**
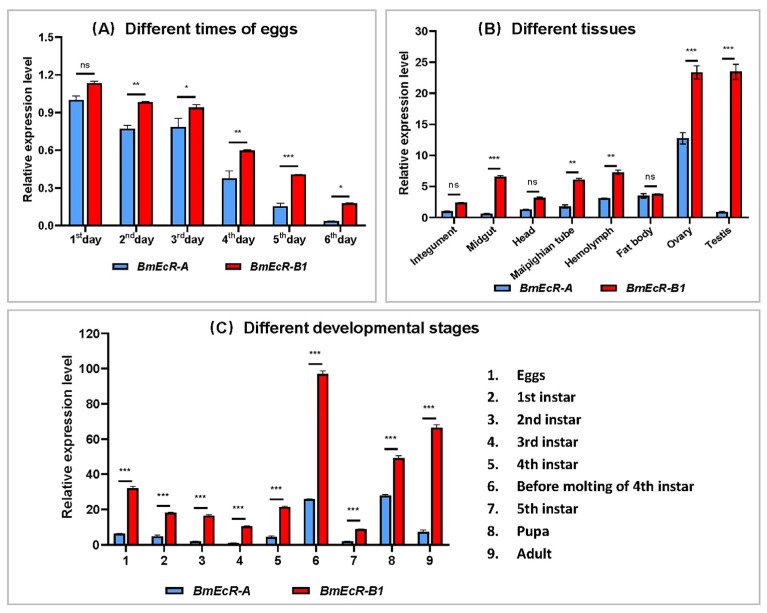
The spatiotemporal expression analysis of *BmEcR-A* and *BmEcR-B1*. *BmEcR-A* and *BmEcR-B1* expression levels at different egg stages (**A**), in different tissues of the third day of fifth instar (**B**), and at different developmental stages (**C**). Although *BmEcR-B1* and *BmEcR-A* shared a similar expression pattern in the spatio-temporal expression spectrums, the levels of *BmEcR-A* are statistically much lower than that of *BmEcR-B1*. All the data were analyzed using Student’s *t*-test and were presented as the mean ± standard error of three independent replicates. Significant differences are indicated by asterisks (*p* < 0.05). ns, no significant; * *p* < 0.05; ** *p* < 0.01; *** *p* < 0.001.

**Figure 2 insects-14-00505-f002:**
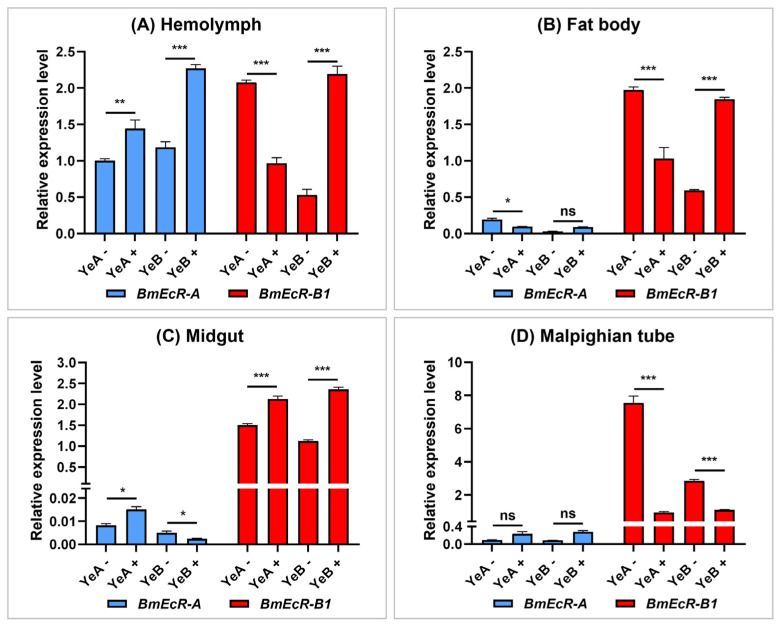
The analysis of *BmEcR-A* and *BmEcR-B1* expression in different tissues of different resistant strains 48 h following BmNPV infection. The expression patterns of *BmEcR-A* and *BmEcR-B1* in the hemolymph (**A**), fat body (**B**), midgut (**C**), and the Malpighian tube (**D**). Silkworm larvae on the first day of fifth instar were infected with BmNPV, and the tissues were collected after 48 h of infection. Results showed that *BmEcR-B1* was more important than *BmEcR-A* in response to BmNPV infection. YeA+ and YeB+ were infected with BmNPV. YeA− and YeB− were not infected with BmNPV. The data were analyzed using Student’s *t*-test and were presented as the mean ± standard error of three independent replicates. Significant differences are indicated by asterisks (*p* < 0.05). ns, no significant; * *p* < 0.05; ** *p* < 0.01; *** *p* < 0.001.

**Figure 3 insects-14-00505-f003:**
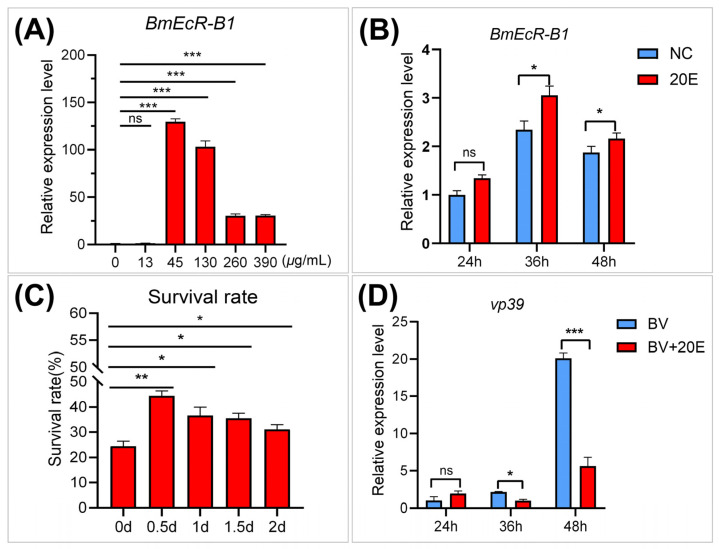
Analysis of the function of 20E on *BmEcR-B1* expression and BmNPV infection. (**A**) *BmEcR-B1* expression after treatment with different doses of 20E. (**B**) Analysis of *BmEcR-B1* expression in silkworm larvae at different times after being fed with 45 μg/mL of 20E. (**C**) The survival rate of BmNPV-infected larvae at different times after being fed with 45 μg/mL of 20E. (**D**) Analysis of *vp39* expression at different times after being fed with 45 μg/mL of 20E. Results showed that 20E improved *BmEcR-B1* expression and inhibited BmNPV infection. (**A**,**B**,**D**) were analyzed using Student’s *t*-test, and (**C**) was the Kruskal Wallis test. NC, negative control. The data were presented as mean ± standard error of three independent replicates. Significant differences are indicated by asterisks (*p* < 0.05). ns, no significant; * *p* < 0.05; ** *p* < 0.01; *** *p* < 0.001.

**Figure 4 insects-14-00505-f004:**
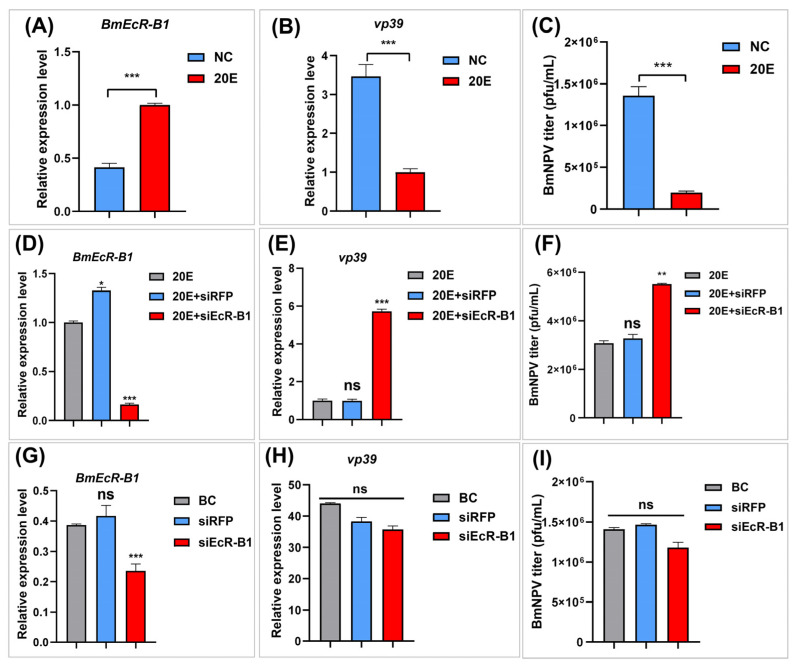
Analysis of the relationship of 20E, *BmEcR-B1*, and BmNPV infection. The expression levels of *BmEcR-B1* (**A**) and *vp39* (**B**) after 48 h of 20E treatment in BmN cells. (**C**) The copy number of BV-eGFP after 48 h of 20E treatment. (**D**) The expression level of *BmEcR-B1* after 48 h of treatment with siRNA and 20E. The optimal time for siEcR-B1 to interfere with *BmEcR-B1* expression was 48 h after transfection (Appendix A). The expression level of *vp39* (**E**) and the copy number of BV-eGFP (**F**) at 48 h after treatment with siRNA and 20E. The expression level of *BmEcR-B1* (**G**), the expression level of *vp39* (**H**), and the copy number of BV-eGFP (**I**) at 48 h post-siRNA transfection in the absence of 20E. BC, blank control. siRFP, negative control (NC). The results showed that *BmEcR-B1* is vital for 20E to inhibit virus proliferation. All the data were analyzed using Student’s *t*-test and were presented as the mean ± standard error of three independent replicates. Significant differences compared to the control and are indicated by asterisks (*p* < 0.05). ns, no significant; * *p* < 0.05; ** *p* < 0.01; *** *p* < 0.001.

**Figure 5 insects-14-00505-f005:**
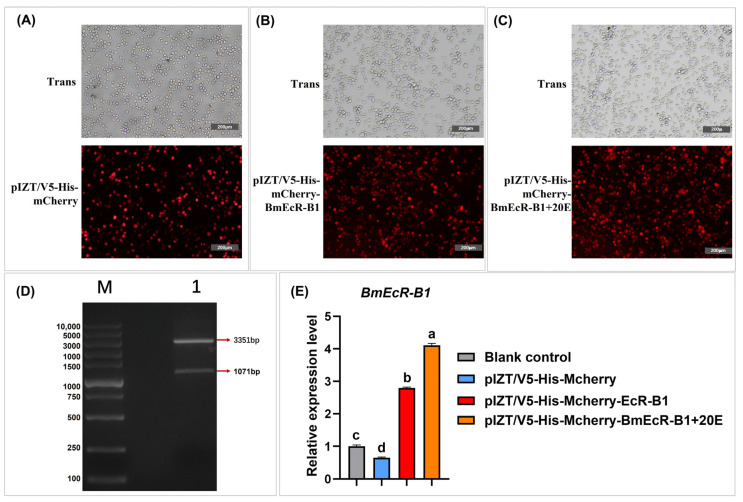
Construction of pIZT/V5-His-mCherry-BmEcR-B1 and detection of *BmEcR-B1* overexpression in BmN cells. The stable cell line of pIZT/V5-His-mCherry (**A**), pIZT/V5-His-mCherry-BmEcR-B1 (**B**), pIZT/V5-His-mCherry-BmEcR-B1 treatment with 20E (**C**). Scale bar = 200 μm. Trans, optical transmission. Red, mCherry. (**D**) Validation of recombinant plasmid pIZT/V5-His-mCherry-BmEcR-B1 with *EcoR*I and *Xba*I. (**E**) The expression level of *BmEcR-B1* in the stable cell line before and after 20E treatment. pIZT/V5-His-mCherry, negative control. The results showed that *BmEcR-B1* was successfully overexpressed in BmN cells using pIZT/V5-His-mCherry. The data were analyzed using Student’s *t*-test and were presented as the mean ± standard error of three independent replicates. Significant differences compared to the control and are indicated by different letters (a, b, c, d).

**Figure 6 insects-14-00505-f006:**
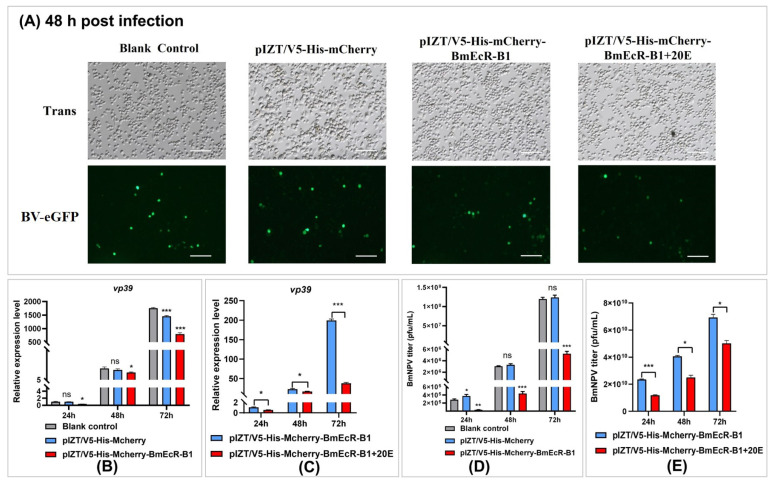
Analysis of the role of *BmEcR-B1* overexpression on BmNPV infection at different times in BmN cells. (**A**) The representative fluorescent pictures of BV-eGFP signal at 48 h post-infection in the stable cell line. Scale bar = 200 μm. Trans, optical transmission. Green, eGFP. pIZT/V5-His-mCherry, negative control. The expression level of *vp39* at 24 h, 48 h, and 72 h post-infection in the stable cell line (**B**) and 20E treatment (**C**). The copy number of BV-eGFP at 24 h, 48 h, and 72 h post-infection in the stable cell line (**D**) and 20E treatment (**E**). The results showed that overexpression of *BmEcR-B1* inhibits viral replication. The data were analyzed using Student’s *t*-test and were presented as the mean ±standard error of three independent replicates. Significant differences compared to the control and are indicated by asterisks (*p* < 0.05). ns, no significant; * *p* < 0.05; ** *p* < 0.01; *** *p* < 0.001.

**Figure 7 insects-14-00505-f007:**
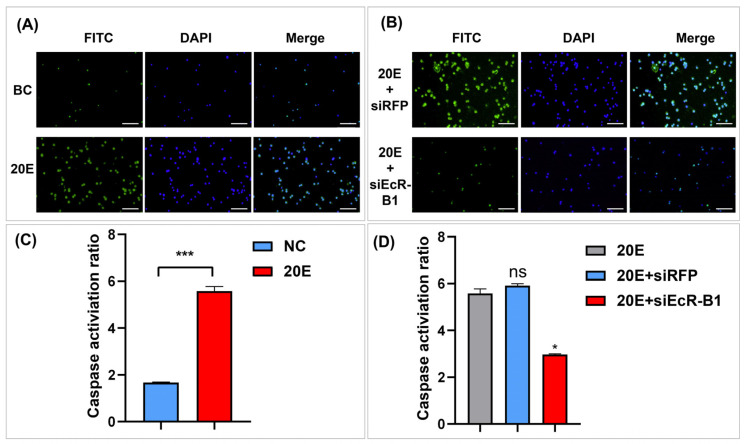
The analysis of the role of *BmEcR-B1* in 20E-induced apoptosis. Detection of apoptotic bodies after treatment with 20E (**A**) and 20E/siEcR-B1 (**B**). DAPI (blue), nuclear dye. FITC (green), apoptotic bodies. Scale bar = 200 μm. The analysis of caspase activities after treatment with 20E (**C**) and 20E/siEcR-B1 (**D**). The results showed that *BmEcR-B1* inhibits viral infection related to the activation of apoptosis. The data were analyzed using Student’s *t*-test and were presented as the mean ± standard error of three independent replicates. Significant differences are indicated by asterisks (*p* < 0.05). ns, no significant; * *p* < 0.05; *** *p* < 0.001.

**Figure 8 insects-14-00505-f008:**
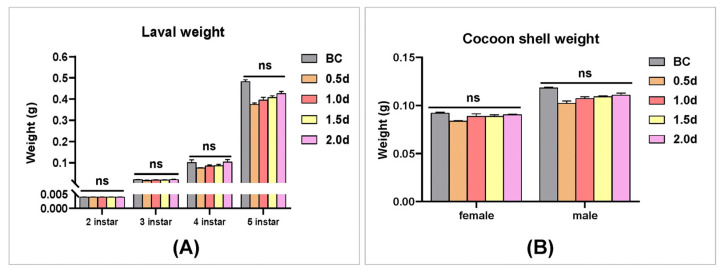
The statistical analysis of silkworm larvae development and cocoon shell after 20E treatment. (**A**) Larval weight of silkworms in different instars after being fed with 45 μg/mL of 20E 0.5 d, 1.0 d, 1.5 d, and 2.0 d after ecdysis. (**B**) Cocoon shell weight of different silkworm genders after being fed with 45 μg/mL of 20E 0.5 d, 1.0 d, 1.5 d, and 2.0 d after ecdysis. BC, blank control, was fed with ddH_2_O. The results showed that 20E treatment did not significantly affect silkworm larvae development and cocoon shell. The data were analyzed using the Kruskal–Wallis test and presented as mean ±standard error of three independent replicates. ns, no significant.

**Figure 9 insects-14-00505-f009:**
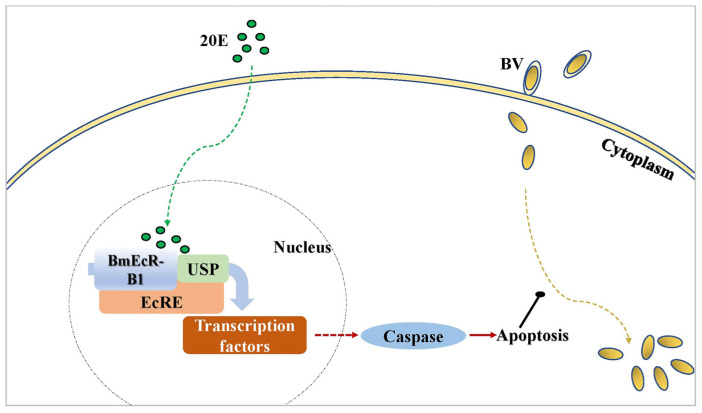
The schematic diagram of the mechanism of *BmEcR-B1* in response to BmNPV infection.

**Table 1 insects-14-00505-t001:** Primers used to amplify gene ORF and RT-qPCR.

Gene Name	Forward Primer (5′-3′)	Reverse Primer (5′-3′)	Note
*BmEcR-B1-O*	CGGAATTCATGGACAGAGCCTCCGGATAC	GCTCTAGAGATGCACATGTTGGAGTTTTG	ORF amplification
*BmEcR-B1*	GCCTCCCACAACACCGAAATCA	TCTGGCGTCAGCATCAGCACT	98.30%, 106 bp
*BmEcR-A*	ACTCGTCGCCACTATCCTCAGG	TGAAGCCGGCGAAAGTTCCT	98.48%, 110 bp
*BmGAPDH*	CCGCGTCCCTGTTGCTAAT	CTGCCTCCTTGACCTTTTGC	116.00%, 98 bp
*vp39*	CAACTTTTTGCGAAACGACTT	GGCTACACCTCCACTTGCTT	100.71%, 125 bp

**Table 2 insects-14-00505-t002:** The list of primer sequences used to synthesize siRNA.

Primer Names	Sequences (5′-3′)
EcR-Olig1-1	GATCACTAATACGACTCACTATAGGGAATTTGGACATGCCTGTGAAATT
EcR-Olig1-2	AATTTCACAGGCATGTCCAAATTCCCTATAGTGAGTCGTATTAGTGATC
EcR-Olig1-3	AAAATTTGGACATGCCTGTGAAACCCTATAGTGAGTCGTATTAGTGATC
EcR-Olig1-4	GATCACTAATACGACTCACTATAGGGTTTCACAGGCATGTCCAAATTTT
EcR-Olig2-1	GATCACTAATACGACTCACTATAGGGAACACGTTGCGAATTTACATCTT
EcR-Olig2-2	AAGATGTAAATTCGCAACGTGTTCCCTATAGTGAGTCGTATTAGTGATC
EcR-Olig2-3	AAAACACGTTGCGAATTTACATCCCCTATAGTGAGTCGTATTAGTGATC
EcR-Olig2-4	GATCACTAATACGACTCACTATAGGGGATGTAAATTCGCAACGTGTTTT
RFP-Olig-1	GATCACTAATACGACTCACTATAGGGGCACCCAGACCATGAGAATTT
RFP-Olig-2	AAATTCTCATGGTCTGGGTGCCCCTATAGTGAGTCGTATTAGTGATC
RFP-Olig-3	AAGCACCCAGACCATGAGAATCCCTATAGTGAGTCGTATTAGTGATC
RFP-Olig-4	GATCACTAATACGACTCACTATAGGGATTCTCATGGTCTGGGTGCTT

## Data Availability

All the datasets in this study can be provided relay on reasonable request.

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
