# Peer review of "Bombyx mori Ecdysone Receptor B1 May Inhibit BmNPV Infection by Triggering Apoptosis"

_insects, 2023, doi:10.3390/insects14060505_

Round 1

Reviewer 1 Report (New Reviewer)

This paper describes  a fascinating and creative set of experiments on the relationship of the B1 isoform of EcR of Bombyx mori to BmNPV infection.  It is rich in data and should appeal to readers who want to control NPV in silkworms and those more generally interested in the function of nuclear receptors in insect development and physiology. The authors even provide data showing that it may be possible to test ecdysone's ability to reduce NPV infection in sericulture. It is a good paper that can be improved by adding some information and by resisting the temptation to overgeneralize from specific results.

One major suggestion for improvement is to make the paper more accessible to scientists who do not work with silkworms. As examples of such changes, the authors could note that the virus infects the larvae, define terms such as ODV, and explain the differences between resistant and susceptible silkworm strains in the methods. I understand that Figure 4 presented data from the YeA and YeB strains but it was not clear to me why the tissue expression studies used p50 instead of YeA or YeB.  I have some knowledge of lepidopterans so there might be other specific jargon that I am missing. I encourage the authors to write to appeal to the widest possible audience.

Another suggestion is to consider if it is necessary to include the bioinformatics analysis, as this information is not used directly to motivate any of the studies reported later in the paper. The presentation of this analysis here means that it is not fully discussed or placed into the full context of the existing literature. 

The description of the methods used to acquire the data presented in Figures 7, 8, and 9 is inadequate. Provide more detail in the methods sections. It is very challenging to interpret the photomicrographs. Maybe several representative photomicrographs can be shown and the remainder of the data presented in the graphs, as only the quantitative analysis is meaningful.

Specific comments and editorial suggestions follow.

1. Title: I have not previously seen the term "ecdysis hormone receptor." The correct term for the nuclear proteins designated EcR is ecdysone receptor. This is important because there is an insect hormone with a similar name, ecdysis triggering hormone.

2. The title makes the findings reported appear more definitive than they are. The pieces of their claim are present in the manuscript, but they have not yet been assembled into full support for an unqualified claim. If I read this paper correctly, apoptosis and its regulation were only demonstrated in the BmN cell line, not in infect larvae.  The authors should use a title that better reflects their data. I suggest Bombyx mori EcR-B1 may inhibit BmNPV infection by triggering apoptosis. The text of the paper itself is more modest and appropriate (for example, lines 48-49 and lines 447-448).

3. Lines 73-74: I don't think it is fair to say that one nuclear receptor is more important than another. I think it is more accurate to say that EcR-B1 has broader functions than EcR-A.

4. In vivo treatments with 20E (Section 2.10). The doses of hormone appear high (supraphysiological). Can the authors relate them to any previously measured titters for Bombyx mori at the developmental stages tested. Also, how can they be certain that each larva received a specific dose by feeding?

5. Line 196: sex is the correct term here (not sexuality).

6. Section 3.1: I think this belongs in another paper because the results provided in this section are not covered in the Discussion section of the manuscript. I be interested to read a fuller discussion of the significance of the absence of the ligand binding domain of the hormone receptor EcR-A in that paper.

7. Line 251: when I look at Figure 3, Panel A, I think that the term almost can be deleted as the early days are reported to have higher levels of expression than the later days of egg development.

8. Line 254: the use of the pronoun them is potentially confusing. Write out BmEcR-A and BmEcR-B1 instead.

9. Line 254: change metamorphosis development stages to during metamorphosis.

10. Line 257 change was much statistically lower than to was significantly lower than.

11. The legend for Figure 3 is incorrect. Panel B is different tissues and Panel C is different developmental stages. Also, it is not stated from what stage the data presented in Panel B were obtained (presumably adult). It would be interesting if these data were discussed in terms of the tissues known to be infected by BmNPV. Were the tissues used to produce the data for Figure 4 from adults?

12. Line 290: should this say the survival rate of infected larvae? 

13. Line 292: this description doesn't seem accurate (most pronounced after 0.5 days), given that there is no difference between any of the days except in comparison to 0 days. Make it clear that the 20E was fed only once. To me it seems more interesting that the effect on survival was apparent regardless of the day of feeding.

14. Section 3.8. How does larval weight and cocoon shell weight of these groups compare with that of the infected larvae treated with 20E (data from Figure 5)? That comparison might provide insight into the use of 20E as a feeding additive in sericulture.

15. Line 397-398: I think it is an exaggeration, based on the evidence, to write that EcR regulates the insect immune response. It contributes to, or modifies, or something else lesser than regulation. Also, reference 22 appears to be irrelevant to the point being made, and should be deleted.

16. Lines 413-414: again, I think important is a term that implies judgment. The functions of the receptors are different, but it is not a competition. 

17: Line 416: change the key gene to a key gene.

18. Suggested edit for title of section 4.3: BmEcR-B1 likely inhibits virus infection in BmN cells via activation of apoptosis.

19. References to EcR-B1associated apoptosis in silk glands and Helicoverpa midgut are included but not fully discussed. Do the authors think their results provide another example of this phenomenon, or do they think that developmental cell death and death of infected cells are different? The possible connection is very interesting.

Author Response

Xue-yang Wang, PhD., Associate Professor

Jiangsu Key Laboratory of Sericutural Biology and Biotechnology, School of Biotechnology, Jiangsu University of Science and Technology

Key Laboratory of Silkworm and Mulberry Genetic Improvement, Ministry of Agricultural and Rural Affairs, Sericultural Research Institute, Chinese Academy of Agricultural Sciences,

Zhenjiang, Jiangsu 212100, China

xueyangwang@just.edu.cn (email)

5th May 2023

Dear Dr. Ivana Vostic and reviewers,

We are resubmitting our revised manuscript entitle “Bombyx mori ecdysone receptor B1 may inhibit BmNPV infection by triggering apoptosis” (Manuscript ID: insects-2337332). We are very grateful for your professional comments and suggestions on our manuscript, and these comments and suggestions are very helpful for us to improve manuscript quality. We have revised the manuscript according to the comments and suggestions of reviewers and editor and responded point by point to the comments as list below. The revised parts were marked in red in the revised manuscript. We greatly appreciate your earnestly editorial work and hope that the revised manuscript could be considered for publication in Insects.

Yours sincerely,

Xue-yang Wang

Response to Reviewer #1:

One major suggestion for improvement is to make the paper more accessible to scientists who do not work with silkworms. As examples of such changes, the authors could note that the virus infects the larvae, define terms such as ODV, and explain the differences between resistant and susceptible silkworm strains in the methods. I understand that Figure 4 presented data from the YeA and YeB strains but it was not clear to me why the tissue expression studies used p50 instead of YeA or YeB. I have some knowledge of lepidopterans so there might be other specific jargon that I am missing. I encourage the authors to write to appeal to the widest possible audience.

Response: p50 strain has been used to build the silkworm genome and maintained in many laboratories. The analysis of expression pattern in p50 will provide better data reference for audience. The description has been added in line 275.

Another suggestion is to consider if it is necessary to include the bioinformatics analysis, as this information is not used directly to motivate any of the studies reported later in the paper. The presentation of this analysis here means that it is not fully discussed or placed into the full context of the existing literature.

Response: The Figure 1 and 2 were moved to supplementary materials.

The description of the methods used to acquire the data presented in Figures 7, 8, and 9 is inadequate. Provide more detail in the methods sections. It is very challenging to interpret the photomicrographs. Maybe several representative photomicrographs can be shown and the remainder of the data presented in the graphs, as only the quantitative analysis is meaningful.

Response: The description of the methods used to acquire Figure 7, 8, and 9 was revised (lines 193, 205, 216). Figure 6 (Original Figure 8) kept representative photomicrographs of 48 h post infection, and the picture has been revised and replaced in the text.

Specific comments and editorial suggestions follow.

Point 1. Title: I have not previously seen the term "ecdysis hormone receptor." The correct term for the nuclear proteins designated EcR is ecdysone receptor. This is important because there is an insect hormone with a similar name, ecdysis triggering hormone.

Response: Thank you for your professional suggestion. All of "ecdysis hormone receptor" have been replaced with ecdysone receptor (lines 2, 13, 23).

Point 2. The title makes the findings reported appear more definitive than they are. The pieces of their claim are present in the manuscript, but they have not yet been assembled into full support for an unqualified claim. If I read this paper correctly, apoptosis and its regulation were only demonstrated in the BmN cell line, not in infect larvae. The authors should use a title that better reflects their data. I suggest Bombyx mori EcR-B1 may inhibit BmNPV infection by triggering apoptosis. The text of the paper itself is more modest and appropriate (for example, lines 48-49 and lines 447-448).

Response: Thank you for your professional suggestion. The title has been revised (Page 1).

Point 3. Lines 73-74: I don't think it is fair to say that one nuclear receptor is more important than another. I think it is more accurate to say that EcR-B1 has broader functions than EcR-A.

Response: Thank you for your professional suggestion. It has been revised (line 74).

Point 4. In vivo treatments with 20E (Section 2.10). The doses of hormone appear high (supraphysiological). Can the authors relate them to any previously measured titters for Bombyx mori at the developmental stages tested. Also, how can they be certain that each larva received a specific dose by feeding?

Response: One of previous studies used 10 mg/mL of 20E to inject silkworms to regulate gene expression (Reference: https://doi.org/10.1111/imb.12288); thus, it should be reasonable to use a dose of 45 mg/mL of 20E to feed silkworm larvae. The description and reference have been added (lines 461-463).

Point 5. Line 196: sex is the correct term here (not sexuality).

Response: It has been revised (line 214).

Point 6. Section 3.1: I think this belongs in another paper because the results provided in this section are not covered in the Discussion section of the manuscript. I be interested to read a fuller discussion of the significance of the absence of the ligand binding domain of the hormone receptor EcR-A in that paper.

Response: Thank you for your professional suggestion. The results in 3.1 have been discussed in the part of Discussion (lines 447-451).

Point 7. Line 251: when I look at Figure 3, Panel A, I think that the term almost can be deleted as the early days are reported to have higher levels of expression than the later days of egg development.

Response: It was deleted.

Point 8. Line 254: the use of the pronoun them is potentially confusing. Write out BmEcR-A and BmEcR-B1 instead.

Response: It has been revised (line 281).

Point 9. Line 254: change metamorphosis development stages to during metamorphosis.

Response: It has been revised (line 282).

Point 10. Line 257 change was much statistically lower than to was significantly lower than.

Response: It has been revised (line 285).

Point 11. The legend for Figure 3 is incorrect. Panel B is different tissues and Panel C is different developmental stages. Also, it is not stated from what stage the data presented in Panel B were obtained (presumably adult). It would be interesting if these data were discussed in terms of the tissues known to be infected by BmNPV. Were the tissues used to produce the data for Figure 4 from adults?

Response: Sorry for the mistake. It has been revised (lines 289-290). The stage is the 3rd day of fifth instar of larvae, which has been added in the legend (line289). The data for Figure 4 is also the 3rd day of fifth instar of larvae, and the description has been added in Figure 4 legend (line 308).

Point 12. Line 290: should this say the survival rate of infected larvae?

Response: It has been revised (line 319).

Point 13. Line 292: this description doesn't seem accurate (most pronounced after 0.5 days), given that there is no difference between any of the days except in comparison to 0 days. Make it clear that the 20E was fed only once. To me it seems more interesting that the effect on survival was apparent regardless of the day of feeding.

Response: Sorry for our misunderstanding description. It has been revised (line 324).

Point 14. Section 3.8. How does larval weight and cocoon shell weight of these groups compare with that of the infected larvae treated with 20E (data from Figure 5)? That comparison might provide insight into the use of 20E as a feeding additive in sericulture.

Response: Virus infection is fatal. Curing virus-infected silkworms will certainly have an adverse effect on weight and cocoon shell weight, and it would certainly be incomprehensible if the result was no adverse effect. To be able to save part of the loss caused by virus disease is already the greatest achievement, and it is very difficult to achieve no effect. Despite the reason, your suggestion is very professional and provides a reference to refine the application of the results of this study.

Point 15. Line 397-398: I think it is an exaggeration, based on the evidence, to write that EcR regulates the insect immune response. It contributes to, or modifies, or something else lesser than regulation. Also, reference 22 appears to be irrelevant to the point being made, and should be deleted.

Response: Thank you for your professional suggestion. The description has been revised (line 432). The reference 22 has been detected.

Point 16. Lines 413-414: again, I think important is a term that implies judgment. The functions of the receptors are different, but it is not a competition.

Response: It has been revised (lines 74, 451).

Point 17. Line 416: change the key gene to a key gene.

Response: It has been revised (line 455).

Point 18. Suggested edit for title of section 4.3: BmEcR-B1 likely inhibits virus infection in BmN cells via activation of apoptosis.

Response: Thank you for your professional suggestion. It has been revised (line 480).

Point 19. References to EcR-B1associated apoptosis in silk glands and Helicoverpa midgut are included but not fully discussed. Do the authors think their results provide another example of this phenomenon, or do they think that developmental cell death and death of infected cells are different? The possible connection is very interesting.

Response: The results of this study provide another example of this phenomenon. The description has been added in line 488.

Reviewer 2 Report (New Reviewer)

The authors present some interesting data suggesting that an ecdysone receptor is associated with immune responses to infection with BmNPV 2. While the data are interesting, there are some flaws that must be addressed.

Major Comments

1.       A previous study that examined the stability of expression of reference genes in fifth instar silkworm up to 24 h after treatment with heat or virus. These results are not applicable to the stability of GAPDH in egg, 1st-4th instar larvae, in Malpighian tubule or hemolymph of 5th instar larvae, to cells transfected with a overexpression plasmid, samples taken 48 h after viral treatment, or in RNAi experiments. Further, MIQUE guidelines recommend 2 or more reference genes for relative qPCR studies unless the stability of a single reference gene is clearly documented under the exact experimental conditions. As it stands, the authors qPCR data is unreliable.

2.       What is the efficiency of amplification for each of the qPCR primer sets? What are the product sizes?

3.       Figure 3: The authors inappropriately compared the transcript levels of EcR-A with EcR-B1. Measurement of these different transcripts uses different sets of primers that each have their own amplification efficiency. This is like comparing apples to oranges and violates statistical assumptions. This figure and the associated text should be rewritten to focus on differences in transcript levels of EcR-A among the different treatments and EcR-B1 among the different treatments. They can say that expression of B1 was higher, though, but higher expression does not necessarily correlate with importance of a gene to an organism’s survival. Also, the authors used ddCt to calculate fold changes in expression, but which condition was set as “normal” in each panel? E.g. which bar was “1” and how was it chosen?

4.       Figure 5C lacks control groups for survival after no treatment and treatment with virus to compare with virus with 20E treatment. What does NC mean? This should be defined in the figure legend of all figures it appears in.

5.       Figure 6, Lines 326-327: The authors state that the data was analyzed by Student’s t-test, a statistical test for determining significant differences between 2 groups, but panels D-I all have three treatment groups. This is also the case for Figure 7.

6.       How were the fluorescent microscopy studies in Figures 7 and 8 performed?

Minor Comments

1.       Lines 48-49 don’t belong here. These are results or may belong at the end of the introduction

2.       Lines 73-74: Basing this conclusion on transcript levels is insufficient

3.       Line 106: What is ODV? Please define.

4.       Line 104: What instar were tissues collected from to analyze the immune response of EcR-A and B1?

5.       Line 155: the two strands were annealed.

6.       Lines 217-221: How was normality of the data assessed?

7.       Lines 146-158: How long are the “siRNAs”? Are they short interfering RNAs or are they long RNAs?

Author Response

Xue-yang Wang, PhD., Associate Professor

Jiangsu Key Laboratory of Sericutural Biology and Biotechnology, School of Biotechnology, Jiangsu University of Science and Technology

Key Laboratory of Silkworm and Mulberry Genetic Improvement, Ministry of Agricultural and Rural Affairs, Sericultural Research Institute, Chinese Academy of Agricultural Sciences,

Zhenjiang, Jiangsu 212100, China

xueyangwang@just.edu.cn (email)

5th May 2023

Dear Dr. Ivana Vostic and reviewers,

We are resubmitting our revised manuscript entitle “Bombyx mori ecdysone receptor B1 may inhibit BmNPV infection by triggering apoptosis” (Manuscript ID: insects-2337332). We are very grateful for your professional comments and suggestions on our manuscript, and these comments and suggestions are very helpful for us to improve manuscript quality. We have revised the manuscript according to the comments and suggestions of reviewers and editor and responded point by point to the comments as list below. The revised parts were marked in red in the revised manuscript. We greatly appreciate your earnestly editorial work and hope that the revised manuscript could be considered for publication in Insects.

Yours sincerely,

Xue-yang Wang

Response to Reviewer #2:

Major Comments

Point 1. A previous study that examined the stability of expression of reference genes in fifth instar silkworm up to 24 h after treatment with heat or virus. These results are not applicable to the stability of GAPDH in egg, 1st-4th instar larvae, in Malpighian tubule or hemolymph of 5th instar larvae, to cells transfected with a overexpression plasmid, samples taken 48 h after viral treatment, or in RNAi experiments. Further, MIQUE guidelines recommend 2 or more reference genes for relative qPCR studies unless the stability of a single reference gene is clearly documented under the exact experimental conditions. As it stands, the authors qPCR data is unreliable.

Response: The optimal reference gene of Bombyx mori glyceraldehyde-3-phosphate dehydrogenase (BmGAPDH) was selected based on the reference that mentioned above. “The results of RNA-seq and qPCR showed that GAPDH and TIF-4A were suitable RGs after BmNPV challenge or HT stress, whereas TIF-4A was an appropriate RG for BmCPV or BmDNV-Z challenge in silkworms”, which is the description of the results of this study. Moreover, it has been widely used in the study of silkworm in response to BmNPV infection (Liu et al., 2023, International Journal of Biological Macromolecules; Zhu et al., 2022, Int. J. Mol. Sci).

Point 2. What is the efficiency of amplification for each of the qPCR primer sets? What are the product sizes?

Response: The efficiency of amplification for each of the qPCR primer sets was added in the table 1, as well as the product sizes.

Point 3. Figure 3: The authors inappropriately compared the transcript levels of EcR-A with EcR-B1. Measurement of these different transcripts uses different sets of primers that each have their own amplification efficiency. This is like comparing apples to oranges and violates statistical assumptions. This figure and the associated text should be rewritten to focus on differences in transcript levels of EcR-A among the different treatments and EcR-B1 among the different treatments. They can say that expression of B1 was higher, though, but higher expression does not necessarily correlate with importance of a gene to an organism’s survival. Also, the authors used ddCt to calculate fold changes in expression, but which condition was set as “normal” in each panel? E.g. which bar was “1” and how was it chosen?

Response: Thank you for your professional comments. One of the purposes of Figure 3 was to determine the expression differences between BmEcR-B1 and BmEcR-A. To achieve this purpose, the amplification efficiency of BmEcR-B1 and BmEcR-A was given special attention. We designed three primer pairs separately and selected primers for two genes with close amplification efficiency for analysis. During the RT-qPCR analysis, both genes were placed on the same 96-well plate for analysis. Thus, the two genes were compared on the same picture. The first set of samples was usually chosen as "1". The description of the relevance and importance of gene expression has been modified (lines74, 451).

Point 4. Figure 5C lacks control groups for survival after no treatment and treatment with virus to compare with virus with 20E treatment. What does NC mean? This should be defined in the figure legend of all figures it appears in.

Response: The variant in Figure 5C is the 20E treatment at different time points, and the results show the differences among the treatments at different time points. The definition of NC was added (line 333).

Point 5. Figure 6, Lines 326-327: The authors state that the data was analyzed by Student’s t-test, a statistical test for determining significant differences between 2 groups, but panels D-I all have three treatment groups. This is also the case for Figure 7.

Response: Three treatment groups in Figure 6 D-I and Figure 7 were compared to the blank control using a comparison of the two data sets. The description has been added in the legends of Figure 6 (line 359), and 7 (line 375), 8 (line 393).

Point 6. How were the fluorescent microscopy studies in Figures 7 and 8 performed?

Response: The captured data of the control group setup is used as the background. The description has been added in line 239.

Minor Comments

Point 1. Lines 48-49 don’t belong here. These are results or may belong at the end of the introduction

Response: It was deleted.

Point 2. Lines 73-74: Basing this conclusion on transcript levels is insufficient

Response: The description has been revised.

Point 3. Line 106: What is ODV? Please define.

Response: It was defined (line 99).

Point 4. Line 104: What instar were tissues collected from to analyze the immune response of EcR-A and B1?

Response: The tissues were collected on the first day of 5th instar larvae. The description has been added in line 106.

Point 5. Line 155: the two strands were annealed.

Response: It has been revised (line 158).

Point 6. Lines 217-221: How was normality of the data assessed?

Response: The normality of the data was assessed with The Kruskal Wallis test in in SPSS software. The description has been revised (line 246).

Point 7. Lines 146-158: How long are the “siRNAs”? Are they short interfering RNAs or are they long RNAs?

Response: The siRNA used in this study was short RNA of 19 bp. It was added in line 152.

This manuscript is a resubmission of an earlier submission. The following is a list of the peer review reports and author responses from that submission.

Round 1

Reviewer 1 Report

Research articule entitled: “Bombyx mori ecdysis hormone receptor B1 inhibits BmNPV infection by triggering apoptosis”

Dear Authors, I read your article with pleasure, the topic itself is very interesting and from my point of view the knowledge conteined in it might be useful in future usage. Although good impression I have few questions and comments.

Lines 67-68: In that lines you are writing about ecdysone and its function in development and physiology wchich is obtained thanks to apoptosis.Unfortunately ecdysteroids  are also the factors which activated genes in epidermal cells which are responsible for synthesis of new cuticule, are elicitors of the deposition of eidermis cuticule and causes stimulation of growth of imaginal discs. So their function is not only results of desintegration ( e.g. apoptosis) but also formation. This information should be mentioned in this paragraph just to nit give the wrong impression or not full knowledge.

Lines 75-77: ”… BmEcR-B1 is expressed predominantly in various tissues”. If something is present in so many tissues along the whole insect body might suggest that their function is very important. So maybe it is worth to stress this in here.

Materials and methods:

Line 93: what was the humidity and light:dark ratio when the final two instars are breed? Add this info.

Line 96: “1% penicillin and streptomycin”. 1% of each antibiotic or in total mix? If that v/v or w/w? Add this info.

Line 102: “A volume of 2.0 μL”. What % of total larvae haemolypmh volume it is? What is the volume of silkworm larve haemolymph? Is volume of 2.0 μL not to much when added in injection? BTW how the injection was made? With syringe with what type of needle? The name of company need to be added.

Line 107 and 115: Here the information about the solvent is given. I am wondering why didn’t you used the physiological saline instead of water? The water as hypotonic liquid might be a reason why the apoptosis occurred.

Line 112: “various developmental stages”. Specify which and why those were chosen.

Line 121: “national center for biotechnology information”. Use capital letters.

Line 126: From that part results that the Neighbor-Joining method of evolution analysis method was chosen ad hoc. Did you used e.g. PROT-TEST or any other program to choose the best method?

Table 1: “The list of primers used in this study” be more precise.

Line 159: That kit was used to synthesize two strands in one reaction or 2 strands were synthesized sepparately and then was hybrydized?

Table 2: There is something wrong with the underlining in sequence EcR-Olig2-3 and EcR-Olig2-4 (two many underlined C or one more G need to me underline) as well as RFP-Olig-1 and RFP-Olig-2 (same as above).

Line 183: How long the transfection last?

Line 193: “each with three repetitions”. Were the repetitions equal?

Line 196: was the day/day time of the mortatily checking constant or just conducted during the whole 3rd instar?

Line 201: As above. This info must be more precise.

Line 203: The number analyzed larvae was thirty. How many cocoons were anayzed then? Also 30? Add info.

Line 222: Add information about the licence for GraphPad Prism.

Statistical analysis: The information about the chosen statisctic and results of it must be addedd in appropriate places in text (whole results section), after each analysis (e.g. writing about the result of analysis or when writing about the statistical importance, as well as information about conducted test must be added under each graph) without it the information is not full and in my opinion irrelevant.

Line 237-238: no capital letter in second word of species name.

Figure 1: This Figure should be also added in supplement, just to give the reader opportunity to look at those data. Now the quality is to low to maximize and look at it more thoroughly.

Figure 2: I cannot see anything on that figure, the background need to be changes from black to any other brighter colour. And second thing, what the colour mean on that graph? Add this info under the tree.

Line 250: so many word “different” in whole text. It needs to be changed as the reader is not supposted to guess but to read it directly from the text.

Line 253: expression level of both…

Line 253: “BmEcR-B1 were much higher in the early developmental stages”. Also the difference between A and B1 is also visible – in case of B1 is was always higher that A. Please check it statistically and if so add this info.

Figure 4: The + means that it was after infection? Add this info

Line 287: There is a mistake in the paragraph name, the dot is in wrong place

Line 296: “the most excellent effect” – two stars is IMHo not excellent aside from the fact that the word biologically means nothing.

Figure 5: Here we see 4 graphs, each of them is prepared in different way: one to indicate the statistical diffrenced the letters are used, then the stars. Also from graphical point of view one the lines are used and one the poliline is used. It needs to be corrected though whole manustript and one of oprion must be chosen. Also on figure 5A: what does the number 1-6 mean? It needs to be added in legend.

Line 348: was the sequence checked in any way or the obtained effects were the indicator of obtaining of proper product?

Line 355: the cinstruct of “confirmed it could” is wrong, if something is confirmed it is taken for granted to if you use a word confirmed the statement shoud sound like: confirmed is is regulated” or if it still is a assumption use different words.

Figure 7: What is the inscription of the lower part of figure 7A? delete it from the figure. Do you have a pictures with DAPI staining in here? It would be better to visualise the cells with nucleus.

Figure 8: The pictures from the microscope are in very low quality, they are indistinct. Also the scale bar is blurry.

Line 390: activity

Figure 9: Pictures need to be changed, as above the quality is very low and I am not sure if these are really apoptotic bodies or not the artefacts and the scale bar is blurry.

Line 423: what previous data? Give the cytations.

Line 427: “development stage” and “testis, ovary” there are different things and it needs to be seperated somehow.

Discussion: In my opinion the quality of discusion is low. It is rather summary of obtained result than comparion with the other text, articles, reaserches etc. Also the statement in line 463-465 is unauthorized because some of the results are not stitistically important.

Author Response

Xue-yang Wang, PhD., Associate Professor

Jiangsu Key Laboratory of Sericutural Biology and Biotechnology, School of Biotechnology, Jiangsu University of Science and Technology

Key Laboratory of Silkworm and Mulberry Genetic Improvement, Ministry of Agricultural and Rural Affairs, Sericultural Research Institute, Chinese Academy of Agricultural Sciences,

Zhenjiang, Jiangsu 212100, China

xueyangwang@just.edu.cu (email)

27th February 2023

Dear Dr. Ivana Vostic and reviewers,

We are resubmitting our revised manuscript entitle “Bombyx mori ecdysis hormone receptor B1 inhibits BmNPV infection by triggering apoptosis” (Manuscript ID: insects-2183219). We are very grateful for your professional comments and suggestions on our manuscript, and these comments and suggestions are very helpful for us to improve manuscript quality. We have revised the manuscript according to the comments and suggestions of reviewers and editor and responded point by point to the comments as list below. Meanwhile, we also improved the language with the help of Proof-Reading-Service. The revised parts were marked in red in the revised manuscript. We greatly appreciate your earnestly editorial work and hope that the revised manuscript could be considered for publication in Insects.

Yours sincerely,

Xue-yang Wang

Response to Reviewer 1 Comments

(insects-2183219)

Response to Reviewer #1:

Reviewer 1: Dear Authors, I read your article with pleasure, the topic itself is very interesting and from my point of view the knowledge conteined in it might be useful in future usage. Although good impression I have few questions and comments.

Response: Thank you very much for your professional suggestions on this manuscript. We have improved our manuscript according to your comments. The responses to each comment of the reviewer were shown as follows.

Point 1. Lines 67-68: In that lines you are writing about ecdysone and its function in development and physiology which is obtained thanks to apoptosis. Unfortunately, ecdysteroids are also the factors which activated genes in epidermal cells which are responsible for synthesis of new cuticule, are elicitors of the deposition of eidermis cuticule and causes stimulation of growth of imaginal discs. So their function is not only results of desintegration (e.g. apoptosis) but also formation. This information should be mentioned in this paragraph just to nit give the wrong impression or not full knowledge.

Response 1: Thank you for your professional suggestion. The information has been added in the manuscript (Page 2, Line 67-68).

Reference:

Zhang, B.; Yao, B.; Li, X.; Jing, T.; Zhang, S.; Zou, H.; Zhang, G.; Zou, C. E74 knockdown represses larval development and chitin synthesis in Hyphantria cunea. Pesticide Biochemistry and Physiology 2022, 187, 105216, doi:10.1016/j.pestbp.2022.105216.

Point 2. Lines 75-77:”… BmEcR-B1 is expressed predominantly in various tissues”. If something is present in so many tissues along the whole insect body might suggest that their function is very important. So maybe it is worth to stress this in here.

Response 2: Thank you for your professional suggestion. The description of the important function of BmEcR-B1 has been added in the manuscript. (Page 2, Line 83-85)

Point 3. Line 93: what was the humidity and light:dark ratio when the final two instars are breed? Add this info.

Response 3: The information has been added in the manuscript. (Page 3, Line 96)

Point 4. Line 96: “1% penicillin and streptomycin”. 1% of each antibiotic or in total mix? If that v/v or w/w? Add this info.

Response 4: 1% penicillin and streptomycin are v/v, which has been added in the manuscript (Page 3, Line 99)

Point 5. Line 102: “A volume of 2.0 μL”. What % of total larvae haemolypmh volume it is? What is the volume of silkworm larve haemolymph? Is volume of 2.0 μL not to much when added in injection? BTW how the injection was made? With syringe with what type of needle? The name of company need to be added.

Response 5: 2.0 μL of BV-eGFP (1×108 pfu/mL) has enough number of virus particles to infection silkworm larvae, which also has been proved in our previous study. The injection was made using the processed capillary with needle tip. The description was added in the manuscript (Page 3, Line 106).

Point 6. Line 107 and 115: Here the information about the solvent is given. I am wondering why didn’t you used the physiological saline instead of water? The water as hypotonic liquid might be a reason why the apoptosis occurred.

Response 6: The comment is professional. The treatment of silkworm with OVD is oral, and the ODV is suspended in water, so we chose the water as control.

Point 7. Line 112: “various developmental stages”. Specify which and why those were chosen.

Response 7: The description has been added in the manuscript. (Page 3, Line 115)

Point 8. Line 121: “national center for biotechnology information”. Use capital letters.

Response 8: It has been revised. (Page 3, Line 125)

Point 9. Line 126: From that part results that the Neighbor-Joining method of evolution analysis method was chosen ad hoc. Did you used e.g. PROT-TEST or any other program to choose the best method?

Response 9: The description was added in the manuscript (Page 3, Line 131-132).

Point 10. Table 1: “The list of primers used in this study” be more precise.

Response 10: It has been revised (Page 4, Line 156).

Point 11. Line 159: That kit was used to synthesize two strands in one reaction or 2 strands were synthesized sepparately and then was hybrydized?

Response 11: Sorry for our unclear description, the information has been added in the manuscript. (Page 4, Line 167)

Point 12. Table 2: There is something wrong with the underlining in sequence EcR-Olig2-3 and EcR-Olig2-4 (two many underlined C or one more G need to me underline) as well as RFP-Olig-1 and RFP-Olig-2 (same as above).

Response 12: Sorry for our mistake. It has been revised (Page 4, Table 2).

Point 13. Line 183: How long the transfection last?

Response 13: The transfection lasted 24 hours, and the description was added in the manuscript (Page 5, Line 189).

Point 14. Line 193: “each with three repetitions”. Were the repetitions equal?

Response 14: Yes, the repetitions were equal. It has been revised (Page 5, Line 199).

Point 15. Line 196: was the day/day time of the mortatily checking constant or just conducted during the whole 3rd instar?

Response 15: The checking of mortality was constant every 12 h after infection, and the description was added in the manuscript. (Page 5, Line 203)

Point 16. Line 201: As above. This info must be more precise.

Response 16: This detail description was added in the manuscript. (Page 5, Line 210)

Point 17. Line 203: The number analyzed larvae was thirty. How many cocoons were anayzed then? Also 30? Add info.

Response 17: Yes, the number of cocoons was 30 too, and it was added in line 211, page 5.

Point 18. Line 222: Add information about the licence for GraphPad Prism.

Response 18: The license was provided by our institution, which was still under "group subscription". By the way, we followed the requirement of the "GraphPad Prism license agreement".

Point 19. Statistical analysis: The information about the chosen statisctic and results of it must be addedd in appropriate places in text (whole results section), after each analysis (e.g. writing about the result of analysis or when writing about the statistical importance, as well as information about conducted test must be added under each graph) without it the information is not full and in my opinion irrelevant.

Response 19: The detail descriptions of statistical analysis were added in the legends of Figure 3-10.

Point 20. Line 237-238: no capital letter in second word of species name.

Response 20: It was revised (Page 6, Line 247).

Point 21. Figure 1: This Figure should be also added in supplement, just to give the reader opportunity to look at those data. Now the quality is to low to maximize and look at it more thoroughly.

Response 21: Sorry for our mistake, we re-uploaded Figure 1 with higher quality. (Page 7, Figure 1)

Point 22. Figure 2: I cannot see anything on that figure, the background need to be changes from black to any other brighter colour. And second thing, what the colour mean on that graph? Add this info under the tree.

Response 22: The black background in Figure 2 may be the reason of format, it was revised and re-uploaded. BmEcR-B1 and its homologs are in blue, and BmEcR-A and its homologs are in red. The description was suppled in Figure 2 legend. (Page 8, Line 258-260)

Point 23. Line 250: so many word “different” in whole text. It needs to be changed as the reader is not supposted to guess but to read it directly from the text.

Response 23: Thank you for your professional suggestion. The whole manuscript was checked, and the unproper “different” was revised in precise.

Point 24. Line 253: expression level of both…

Response 24: It has been revised. (Page 8, Line 264)

Point 25. Line 253: “BmEcR-B1 were much higher in the early developmental stages”. Also the difference between A and B1 is also visible – in case of B1 is was always higher that A. Please check it statistically and if so add this info.

Response 25: We confirmed that BmEcR-B1 was always statistically higher that BmEcR-A. The information was added in line 270.

Point 26. Figure 4: The + means that it was after infection? Add this info

Response 26: The information has been added to the legend of Figure 4. (Page 10, Line 292)

Point 27. Line 287: There is a mistake in the paragraph name, the dot is in wrong place

Response 27: It has been revised.

Point 28. Line 296: “the most excellent effect” – two stars is IMHo not excellent aside from the fact that the word biologically means nothing.

Response 28: It has been revised. (Page 10, Line 306)

Point 29. Figure 5: Here we see 4 graphs, each of them is prepared in different way: one to indicate the statistical diffrenced the letters are used, then the stars. Also from graphical point of view one the lines are used and one the poliline is used. It needs to be corrected though whole manustript and one of oprion must be chosen. Also on figure 5A: what does the number 1-6 mean? It needs to be added in legend.

Response 29: Figure 5 was made again and re-uploaded, and its legend was also revised (Page 11, Figure 5).

Point 30. Line 348: was the sequence checked in any way or the obtained effects were the indicator of obtaining of proper product?

Response 30: Sorry for our uncomplete description. The sequence was checked by sequencing in Sangong Biotech (Shanghai, China), which was added in line 180.

Point 31. Line 355: the cinstruct of “confirmed it could” is wrong, if something is confirmed it is taken for granted to if you use a word confirmed the statement shoud sound like: confirmed it is regulated” or if it still is an assumption use different words.

Response 31: Thank you for your professional suggestion. It has been revised. (Page 12, Line 365)

Point 32. Figure 7: What is the inscription of the lower part of figure 7A? delete it from the figure. Do you have a pictures with DAPI staining in here? It would be better to visualise the cells with nucleus.

Response 32: Sorry for our mistake, the inscription is a scale bar and we have revised and re-upload Figure 7. We are sorry for the lack of DAPI staining. The red of mCherry is also clear, so it will be easy to visualize (Page 13, Figure 7).

Point 33. Figure 8: The pictures from the microscope are in very low quality, they are indistinct. Also the scale bar is blurry.

Response 33: Figure 8 was made again and re-upload (Page 14, Figure 8).

Point 34. Line 390: activity

Response 34: It has been revised. (Page 14, Line 400)

Point 35. Figure 9: Pictures need to be changed, as above the quality is very low and I am not sure if these are really apoptotic bodies or not the artefacts and the scale bar is blurry.

Response 35: Figure 9 was made again and re-upload. The scale bar looks grey because the image has shrunk and remains white after zooming in (Page 15, Figure 9).

Point 36. Line 423: what previous data? Give the cytations.

Response 36: Sorry for our mistake, we have added the refence [29]: Wang, X. Y., Yu, H. Z., Geng, L., Xu, J. P., Yu, D., Zhang, S. Z., Ma, Y., & Fei, D. Q. (2016). Comparative Transcriptome Analysis of Bombyx mori (Lepidoptera) Larval Midgut Response to BmNPV in Susceptible and Near-Isogenic Resistant Strains. PLoS ONE, 11(5), e0155341. https://doi.org/10.1371/journal.pone.0155341

Point 37. Line 427: “development stage” and “testis, ovary” there are different things and it needs to be seperated somehow.

Response 37: It has been revised. (Page 16, Line 440)

Point 38. Discussion: In my opinion the quality of discusion is low. It is rather summary of obtained result than comparion with the other text, articles, reaserches etc. Also the statement in line 463-465 is unauthorized because some of the results are not stitistically important.

Response 38: We have revised the Discussion. We first adjusted the logic of the discussion, and further summarized the results of the research in each part. Based on discussing the role of BmEcR-B1 in BmNPV infection, we clarified the role of 20E in this process, and finally explored whether this process is related to apoptosis.

Reviewer 2 Report

The article entitled “Bombyx mori ecdysis hormone receptor B1 inhibits BmNPV infection by triggering apoptosis” by  Zhi-hao Su et al. presents data on the antiviral function of 20E with the involvement of its specific receptor BmEcR-B1 and the apoptosis pathway in response to BmNPV infection. By contacting BmEcR-B1 RNAi and overexpression experiments they showed a good correlation either to the infection capacity of BmNPV or to the inhibition of baculovirus infection, respectively, upon ecdysone feeding in larvae or ecdysone treatment of cell cultures. Potentially these results could attribute to the elucidation of the mechanism by which insect hosts combat baculovirus infection via ecdysone mediated apoptosis.

 I found it very difficult to read through this manuscript. There is not a good flow in the text which has a lot of language mistakes and missing logic in several sentences. For instance, a) text as the following:  “As two types of BmNPV, budded virus (BV) and occlusion-derived virus (ODV), are easily obtained and purified. Silkworm larvae are moderate-size with clear genetic backgrounds [3,4].”  is meaningless and confusing. b) Statements such as: “ Bombyx mori ecdysis hormone receptor B1 (BmEcR-B1) was found to respond to BmNPV in our transcriptome data” are not correct since the authors do not present any transcriptome analysis etc…

The introduction is poorly written without a thorough analysis of the cited  literature while in a few cases that I checked, the reviewed statements in the text were accompanied by incorrect references (e.g. references in paragraph 3-- Ref 17: is about PCD in metamorphosis and not expression pattern of the two receptors etc). In the Results section, the description and the rationalization of the experiments is minimal presented in an unacceptable way and in many cases more information was included in the Figure legends than in the main text. Thus it is difficult for the reader to follow the experimental set up (for instance section 3.4) so to understand the value of the result.

Although some of the experiments show quite interesting results, I cannot accept this paper for publication in its current form. I think that the authors should perform a major and careful rewriting taking extra care of the language. More importantly they should present their results with the appropriate rationalization and discussion comments to guide the reviewer through their research and clearly express the points they want to make in combination with the cited literature.

Author Response

Xue-yang Wang, PhD., Associate Professor

Jiangsu Key Laboratory of Sericutural Biology and Biotechnology, School of Biotechnology, Jiangsu University of Science and Technology

Key Laboratory of Silkworm and Mulberry Genetic Improvement, Ministry of Agricultural and Rural Affairs, Sericultural Research Institute, Chinese Academy of Agricultural Sciences,

Zhenjiang, Jiangsu 212100, China

xueyangwang@just.edu.cu (email)

27th February 2023

Dear Dr. Ivana Vostic and reviewers,

We are resubmitting our revised manuscript entitle “Bombyx mori ecdysis hormone receptor B1 inhibits BmNPV infection by triggering apoptosis” (Manuscript ID: insects-2183219). We are very grateful for your professional comments and suggestions on our manuscript, and these comments and suggestions are very helpful for us to improve manuscript quality. We have revised the manuscript according to the comments and suggestions of reviewers and editor and responded point by point to the comments as list below. Meanwhile, we also improved the language with the help of Proof-Reading-Service. The revised parts were marked in red in the revised manuscript. We greatly appreciate your earnestly editorial work and hope that the revised manuscript could be considered for publication in Insects.

Yours sincerely,

Xue-yang Wang

Response to Reviewer 1 Comments

(insects-2183219)

Response to Reviewer #2:

Reviewer 2: The article entitled “Bombyx mori ecdysis hormone receptor B1 inhibits BmNPV infection by triggering apoptosis” by Zhi-hao Su et al. presents data on the antiviral function of 20E with the involvement of its specific receptor BmEcR-B1 and the apoptosis pathway in response to BmNPV infection. By contacting BmEcR-B1 RNAi and overexpression experiments they showed a good correlation either to the infection capacity of BmNPV or to the inhibition of baculovirus infection, respectively, upon ecdysone feeding in larvae or ecdysone treatment of cell cultures. Potentially these results could attribute to the elucidation of the mechanism by which insect hosts combat baculovirus infection via ecdysone mediated apoptosis.

Response: Thank you very much for your professional suggestions. We have seriously revised the manuscript, especially the logic of the Introduction, Results and Discussion. Meanwhile, we also have sent our manuscript to "Proof-Reading-Service" for English editing. The responses to each comment of the reviewer were shown as follows.

Point 1. I found it very difficult to read through this manuscript. There is not a good flow in the text which has a lot of language mistakes and missing logic in several sentences. For instance, a) text as the following: “As two types of BmNPV, budded virus (BV) and occlusion-derived virus (ODV), are easily obtained and purified. Silkworm larvae are moderate-size with clear genetic backgrounds [3,4].”  is meaningless and confusing. b) Statements such as: “Bombyx mori ecdysis hormone receptor B1 (BmEcR-B1) was found to respond to BmNPV in our transcriptome data” are not correct since the authors do not present any transcriptome analysis etc…

Response 1: We have seriously revised our manuscript. (a) we have rewritten the text of the first paragraph of the Introduction. The significance of studying the relationship between the silkworm and baculovirus is clarified, which is of great research value for the improvement of BmNPV resistance of silkworms or the biological control of Lepidoptera pests. (Page 1-2, Line 39-47). (b) we have added the refence [29]:Wang, X. Y., Yu, H. Z., Geng, L., Xu, J. P., Yu, D., Zhang, S. Z., Ma, Y., & Fei, D. Q. (2016). Comparative Transcriptome Analysis of Bombyx mori (Lepidoptera) Larval Midgut Response to BmNPV in Susceptible and Near-Isogenic Resistant Strains. PLoS ONE, 11(5), e0155341 (Page 16, Line 432).

Point 2. The introduction is poorly written without a thorough analysis of the cited literature while in a few cases that I checked, the reviewed statements in the text were accompanied by incorrect references (e.g. references in paragraph 3-- Ref 17: is about PCD in metamorphosis and not expression pattern of the two receptors etc). In the Results section, the description and the rationalization of the experiments is minimal presented in an unacceptable way and in many cases more information was included in the Figure legends than in the main text. Thus, it is difficult for the reader to follow the experimental set up (for instance section 3.4) so to understand the value of the result.

Response 2: Sorry for our mistake, we have seriously revised the manuscript and checked the reference. (Page 2, Line 73) We also enriched the details in the Results.

References:

Kamimura, M.; Tomita, S.; Kiuchi, M.; Fujiwara, H. Tissue-specific and stage-specific expression of two silkworm ecdysone receptor isoforms -- ecdysteroid-dependent transcription in cultured anterior silk glands. Eur J Biochem 1997, 248, 786-793, doi:10.1111/j.1432-1033.1997.t01-1-00786.x.

Point 3. Although some of the experiments show quite interesting results, I cannot accept this paper for publication in its current form. I think that the authors should perform a major and careful rewriting taking extra care of the language. More importantly they should present their results with the appropriate rationalization and discussion comments to guide the reviewer through their research and clearly express the points they want to make in combination with the cited literature.

Response 3: We also have sent our manuscript to "Proof-Reading-Service" for English editing. The responses to each comment of the reviewer were shown as follows. Discussion was revised by summarizing the results in this study and discussing it with other literatures. We first adjusted the logic of the discussion, and further summarized the results of the research in each part. Based on discussing the role of BmEcR-B1 in BmNPV infection, we clarified the role of 20E in this process, and finally explored whether this process is related to apoptosis.

Round 2

Reviewer 1 Report

Thank you for your revision.